# GAME-THEORETIC UNDERSTANDING OF MISCLASSIFICATION

## ABSTRACT

This paper analyzes various types of image misclassification from a game-theoretic view. Particularly, we consider the misclassification of clean, adversarial, and corrupted images and characterize it through the distribution of multi-order interactions. We discover that the distribution of multi-order interactions varies across the types of misclassification. For example, misclassified adversarial images have a higher strength of high-order interactions than correctly classified clean images, which indicates that adversarial perturbations create spurious features that arise from complex cooperation between pixels. By contrast, misclassified corrupted images have a lower strength of low-order interactions than correctly classified clean images, which indicates that corruptions break the local cooperation between pixels. We also provide the first analysis of Vision Transformers using interactions. We found that Vision Transformers show a different tendency in the distribution of interactions from that in CNNs, and this implies that they exploit the features that CNNs do not use for the prediction. Our study demonstrates that the recent game-theoretic analysis of deep learning models can be broadened to analyze various malfunctions of deep learning models including Vision Transformers by using the distribution, order, and sign of interactions.

## 1 INTRODUCTION

Deep learning models misclassify images for various reasons. They fail to classify some clean images in a dataset, and they also misclassify because of adversarial perturbations and common corruption. Understanding the causes of misclassifications in deep learning models is vital for their safe applications in society. Several recent studies provided new directions for understanding deep learning models from game-theoretic viewpoints (Cheng et al., 2021; Deng et al., 2022; Ren et al., 2021; Wang et al., 2021; Zhang et al., 2021). For example, adversarial images (Goodfellow et al., 2015; Szegedy et al., 2014)—the images that are slightly but maliciously perturbed to fool deep learning models—were characterized using the *interaction* (Ren et al., 2021; Wang et al., 2021), which is originally used as a measure of the synergy of two players in game theory (Grabisch & Roubens, 1999). In image classification, the average interaction $I$ of an image is a measure of the average change in the confidence score (i.e., the softmax value of logits of the true class) by the cooperation of various pairs of pixels.

$$I = \mathbb{E}_{(i,j)}[I(i,j)].$$

Here, $I(i,j)$ denotes the interaction of the $i$- and $j$-th pixels, which is, roughly speaking, defined by the difference in their contributions to the confidence score:

$$I(i,j) = g(i,j) - g(i) - g(j) + \text{const.},$$

where $g(k)$ relates to the contribution of the $k$-th pixel to the confidence score. The formal definition of $I(i,j)$ will be provided later in this paper.

When $I(i,j) \approx 0$ for a pixel pair, it indicates that the two pixels contribute to the model prediction almost independently. By contrast, a large interaction indicates that the combination of pixels (e.g., edges) contributes to the model prediction in synergy. An interaction can be presented as an average of interactions of different orders, $I(i,j) = \frac{1}{n-1} \sum_s I^{(s)}$, where $I^{(s)}$ and $n$ denote the interaction of order $s$ and the number of the pixels, respectively. The decomposition into $\left\{ I^{(s)} \right\}_s$ gives a more detailed view of the cooperation of pixels; low-order interactions measure simple cooperation between

pixels, whereas high-order interactions measure relatively global and complex concepts. In other words, low- and high-order interactions correspond to different categories of features. Cheng et al. (2021) investigated the link between the order of interactions and the image features. They showed that in general, low-order interactions reflect local shapes and textures, whereas high-order interactions reflect global shapes and textures that frequently appear in training samples. Ren et al. (2021) showed that adversarial perturbations affect high-order interactions and adversarially trained models are robust to perturbations to the features related to high-order interactions. Zhang et al. (2021) found that the dropout regularizes the low-order interactions. These results suggest that interactions can characterize how deep learning models view images; thus, one can obtain a deeper understanding of the cause of the model predictions through interactions.

In this study, we investigate one of the most fundamental issues of deep learning models, misclassification, through the lens of interactions. We examine various types of misclassifications; we consider misclassification of clean, adversarial, and corrupted images, and characterize them by the distribution, order, and sign of the interactions. In the experiments, we contrasted the distribution of interactions of misclassified images to that of successfully classified clean images, thereby revealing which types of features are more exploited to make a prediction for each set of images. The results show that these three types of misclassifications have a distinct tendency in interactions, which indicates that each of them arises from different causes. The summarized results are as follows:

**Misclassification of clean images.**  The distributions of interactions did not present a large difference between misclassified clean images and those successfully classified. This result indicates that the misclassification is not triggered by distracting the model from the useful features, and the model relies on similar features in images regardless of the correctness of the prediction.

**Misclassification of adversarial images.**  We observed a sharp increase in the strength of interactions in high order, indicating that for adversarial images, the model exploits more features that relate to interactions in these orders. Namely, the misclassification is triggered by the destruction of the model from the useful cooperation of pixels in low order to the spurious one in other orders.

**Misclassification of corruption images.**  We observed that while the interactions moderately increased in high order, they also decrease in low- and middle-order interactions. This indicates that the model can no longer use the originally observed pixel cooperations because of the corruption and gain useless or even harmful pixel cooperations in high order to make a prediction.

The abovementioned results were observed on convolutional neural networks (CNNs; He2). We investigated whether these results generalize to Vision Transformers, which recently shows a better performance in image recognition tasks over CNNs. For DeiT-Ti (Touvron et al., 2021) and Swin-T Transformer (Liu et al., 2021), most results hold but with stronger contrast. Besides, for the misclassification of clean images, where CNNs showed no particular difference in the distribution of the interactions between correctly classified and misclassified images, the Vision Transformers showed a striking difference. This suggests that the characteristics of the predictions are more clearly exhibited for Vision Transformers than CNNs, and thus, the analysis with interactions can be exploited even after the shift from CNNs to Vision Transformers.

We also conducted experiments on adversarial attacks and transferability as they have been discussed extensively in the literature (Croce & Hein, 2020; Dong et al., 2018; Ilyas et al., 2019; Wang et al., 2021; Yang et al., 2021). We discovered that when images are adversarially perturbed, the distribution of interactions shifts to negative values. This reveals that adversarial perturbations break the features that the model exploits or even alter them to misleading ones. We also discovered that the adversarial transferability depends on the order of interaction; adversarial images with higher interactions in high order transfer better when adversarially perturbed using ResNet-18, whereas, interestingly, they transfer less when Swin-T is used. This contrastive tendency is analogous to the recent observations that the adversarial images are more perturbed in the high-frequency domain with CNNs and in the low-frequency domain with Vision Transformers (Kim & Lee, 2022).

The contributions of our study can be summarized as follows:

- This study investigates various types of misclassifications from a game-theoretic perspective for the first time. In particular, we characterize the misclassification of clean, adversarial, and corrupted images with the distribution, order, and sign of the interactions.

- We discover that the three types of misclassifications have different tendencies in interactions, which indicates that each type of misclassification is triggered by different causes. Particularly, the dominant order of interactions suggests which category of features affects the prediction more, which is different between the three types of misclassification.

- We provide the first analysis of Vision Transformers by using interactions and found that the difference in distributions of interactions between misclassified and correctly classified images are clearer and also different from the case with CNNs. We also find that the images that are more adversarially transferable have the opposite tendency in the interactions between Vision Transformers and CNNs.

## 2 RELATED WORK

Recent studies have shown that interactions give a new perspective to understanding the machine vision of deep learning models (Cheng et al., 2021; Deng et al., 2022; Ren et al., 2021; Wang et al., 2021; Zhang et al., 2021). For example, Wang et al. (2021) showed that the transferability of adversarial images has a negative correlation to the interactions and provided a unified explanation of the effectiveness of various methods proposed to enhance the transferability. Ren et al. (2021) also characterized adversarial images using interactions. They showed that adversarial perturbations decrease the high-order interactions and the adversarially trained models are robust against such changes. The low-order and high-order interactions are considered to correspond to different types of image features. This was investigated by (Cheng et al., 2021). Generally, low-order interactions reflect local shapes and textures that frequently appear in training samples, while high-order interactions reflect global shapes and textures in training samples. Furthermore, Deng et al. (2022) argued that humans gain more information than deep learning models when a given image is moderately masked, while it is the opposite when the image is masked much more or less. Interactions also relate to dropout regularization. Zhang et al. (2021) showed the similarity between the computation of interactions (particularly, a variant of the Banzhaf values) and dropout regularization. In addition, they showed that the dropout regularizes the low-order interactions and argued that this is why it leads to a better generalization error when a model equips it.

The abovementioned studies have considered what features in clean images are focused on by deep learning models and/or how the interactions are changed by adversarial perturbations. By contrast, our study newly discusses and characterizes misclassification using interactions. Particularly, we focus on *misclassified* images in noise-free, adversarially perturbed, and corrupted cases. Unlike the existing study, we also consider Vision Transformers, which are known to use low-frequency and shape-related features more than CNNs and thus expected to have a different nature in interactions.

## 3 PRELIMINARIES

**Shapley value.** *Shapley value* was proposed in game theory to measure the contribution of each player to the total reward that is obtained by multiple players working cooperatively (Shapley, 1953), thereby fairly distributing the reward to each player. Let $N = \{1, 2, \ldots, n\}$ be the set of $n$ players. We denote the power set of $N$ by $2^N \overset{\text{def}}{=} \{S \mid S \subseteq N\}$. Let $f : 2^N \to \mathbb{R}$ be a reward function. Then, the Shapley value of player $i$ with a *context* $N$, $\phi(i \mid N)$, is defined as follows.

$$\phi(i \mid N) \overset{\text{def}}{=} \sum_{S \subseteq N \setminus \{i\}} \frac{|S|!(n - |S| - 1)!}{n!} (f(S \cup \{i\}) - f(S)),$$

where $|\cdot|$ denotes the cardinality of set. As the definition demonstrates, $\phi(i \mid N)$ considers the change in the reward when player $i$ joins a set of players $S \subseteq N \setminus \{i\}$ and averages it over all $S$.

**Interaction.** Interaction measures the contribution made by the corporation of two players to the total reward. The following defines the interaction between $i$- and $j$-th players.

$$I(i, j) \overset{\text{def}}{=} \phi(S_{ij} \mid N') - (\phi(i \mid N \setminus \{j\}) + \phi(j \mid N \setminus \{i\})),$$

where two players $i, j$ are regarded as a single player $S_{ij} = \{i, j\}$ and $N' = N \setminus \{i, j\} \cup \{S_{ij}\}$ (note that $|N'| = n - 1$). The first term $\phi(S_{ij} \mid N')$ corresponds to the cooperative contribution of the

two players, whereas the second and the third terms, $\phi(i \mid N \setminus \{j\})$ and $\phi(j \mid N \setminus \{i\})$, correspond to the individual contributions of players $i$ and $j$, respectively. Next, we introduce $I^{(s)}(i,j)$, which is the interaction between players $i, j$ of order $s$.

$$I^{(s)}(i,j) \stackrel{\text{def}}{=} \mathbb{E}_{S \subseteq N \setminus \{i,j\}, |S|=s} \left[ \Delta f(i,j,S) \right],$$

where $\Delta f(i,j,S) \stackrel{\text{def}}{=} f(S \cup \{i,j\}) - f(S \cup \{j\}) - f(S \cup \{i\}) + f(S)$. Note that the size of the context $S$ is fixed to $s$ in the calculation of $I^{(s)}(i,j)$. A low-order (high-order) interaction measures the impact of the cooperation between two players when a small (large) number of players join the game. The average of $I^{(s)}(i,j)$ over $s = 0, 1, \ldots, n-2$ satisfies $I(i,j) = \frac{1}{n-1} \sum_{s=0}^{n-2} I^{(s)}(i,j)$.

## 4 Interpretation of interaction

Here, we describe how the interaction is defined in the image domain and how it can be interpretd.

### 4.1 Interaction in image domain

Image $x$ is considered as a set of players $N$, where each pixel is viewed as a player. The reward function $f(\cdot)$ can be defined by any scalar function based on the output of a deep learning model. In our study, we consider a standard multi-class classification and define the reward function as $f(x) = \log \frac{P(y|x)}{1-P(y|x)}$, where $x$ and $y$ denote the input image and its class, respectively, and $P(y \mid x)$ denotes the confidence score of $y$ given by the model with input $x$. Here, $S \subset N$ corresponds to an image whose pixels not included in $S$ are all masked, and $f(S)$ is the confidence score of $y$ given by the model with this masked image. This choice of the reward function is also adopted in (Deng et al., 2022). With this reward function, Shapley value $\phi(i \mid N)$ represents the average contribution of the $i$-th pixel to the network output. Interaction $I(i,j)$ represents cooperation between the $i$- and $j$-th pixels. The value $\Delta f(i,j,S)$ can be interpreted as the cooperative contribution of the $i$- and $j$-th pixels with context $S$ to the confidence score. For example, $\Delta f(i,j,S) \approx 0$ indicates that the two pixels contribute to the confidence score almost independently, whereas $\Delta f(i,j,S) \gg 0$ ($\Delta f(i,j,S) \ll 0$) indicates that the two pixels cooperatively increase (decrease) the confidence score. Zhang et al. (2021) visualized that the pixel pairs with strong interactions in facial images cluster around the face parts, which demonstrates that the pixels that contribute to the model prediction do not work independently but in synergy.

### 4.2 Order of interaction

Order $s$ of the interactions $I^{(s)}(i,j)$ represents the size of context $S$, where $S$ can be any subset of $N \setminus \{i,j\}$ of size $s$. A multi-order interaction reflects the cooperation of pixels in certain complexity. When $s$ is small, $I^{(s)}(i,j)$ reflects the cooperation between the $i$ and $j$-th pixels without considering most of the other pixels, whereas when $s$ is large, the cooperation between the $i$ and $j$-th pixels is measured by considering most other pixels. Cheng et al. (2021) investigated how the multi-order interactions reflect the shape and texture of images, arguing that low-order interactions reflect local colors and shapes that frequently appear in the training samples while high-order interactions reflect the global textures and shapes that frequently appear in the training samples.

### 4.3 Sign of interaction

The sign of the interactions also has important implications, although most existing studies did not consider this and only consider the strength (i.e., absolute value) of interactions. For example, $\Delta f(i,j,S) > 0$ indicates that the cooperation between $i$- and $j$-th pixels (with context $S$) increases the confidence score, whereas $\Delta f(i,j,S) < 0$ indicates that the cooperation decreases it. Therefore, we can consider that misclassification occurs when $\Delta f(i,j,S)$ are negative for most $(i,j,S)$. In fact, as we will show later in the experiment, the distribution of $\Delta f(i,j,S)$ (for various $(i,j,S)$ triplets) shifts to negative values after adversarial perturbations (cf. Fig. 4). Although the perturbations are very small, the shift is large, indicating that adversarial perturbations efficiently break or even deteriorate the cooperation between pixels.

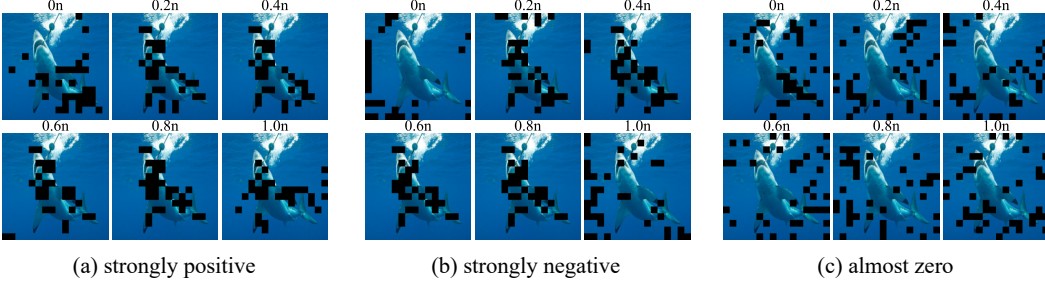

(a) strongly positive  (b) strongly negative  (c) almost zero

Figure 1: Visualization of pixel pairs (marked in black) with (a) strongly positive, (b) strongly negative, and (c) almost zero values for $\Delta f(i, j, S)$ for different orders.

Figure 1 shows the pixel pairs (marked in black) with interactions that are (a) large positive values, (b) large negative values, and (c) almost zero values. In Fig. 1(a), most of the marked pixels cluster around the shark body, indicating that the object's shape and texture closely relate to the shark label and increase the confidence score. In Fig. 1(b), the marked pixels also cluster around the shark body in the middle-order case, implying that those pixel pairs do not work cooperatively to increase the confidence score. We consider that this is because of an overfitting of the model to the non-generalizable features in training samples, which has a negative effect on the prediction. In Fig. 1(c) most pixels distribute over the sea (i.e., background), indicating that the cooperation between these pixels does not directly relate to the shark label.

## 5 INTERACTION ANALYSIS OF VARIOUS TYPES OF MISCLASSIFICATION

In this section, we analyze three types of misclassification from the perspective of interactions. We consider three types of images that are commonly misclassified.

**Clean images.** Typical clean images are unperturbed and they present objects to be classified clearly. Nevertheless, deep learning models can misclassify them because the models do not learn the class-discriminative features that generalize perfectly. Namely, the misclassification of clean images occurs because of the overfitting and underfitting to training samples. Our analysis considers the misclassification of clean images since this is one of the most fundamental issues of image recognition tasks.

**Adversarial images.** Adversarial images are the images slightly but maliciously perturbed to fool deep learning models. Although the perturbations are almost imperceptible to humans, they greatly impact the prediction by deep learning models. This shows the discrepancy between human and machine vision and has been studied extensively (Buckner, 2020; Ilyas et al., 2019; Tsipras et al., 2019). To give further insight into the counter-intuitive misclassification that adversarial images cause, we include them in our analysis.

**Corrupted images.** In contrast to adversarial perturbations, common corruptions (e.g., *fog*, *brightness*, *Gaussian noise*, etc.) occur more commonly in standard settings. Interestingly, adversarially robust models are not robust against some common corruptions (e.g., *fog* and *brightness*) and even perform worse than normally trained models (Yin et al., 2019). In other words, the robustness against adversarial perturbations and common corruptions relates to different nature types in the classifiers, which motivated us to include corrupted images in our analysis. Note that obtaining a robust classifier against both of them—or achieving general robustness—is an unresolved challenge that is addressed in recent studies (Laugros et al., 2019; 2020; Tan et al., 2022).

In particular, we investigate misclassified ones among these images through the distribution, order, and sign of interactions, thereby elucidating what kind of features (in a high-level sense) the deep learning models exploit and make them give incorrect predictions.

**Setup.** We compare the distribution of interactions of misclassified clean, adversarial, and corrupted images with that of successfully classified clean images. Here, the distribution of interactions means the distribution of $\Delta f(i, j, S)$ for various $(i, j, S)$ and images[1]. As the order of interactions grasps the high-level category of image features, we compute the distributions of interactions for different orders. As in Fig. 2, we plot the second quartile (i.e., median) by using a solid line with a shade covering the range between the first and third quartiles. In the plot, the result for the correctly classified images is also superposed for comparison. Intuitively, the second quartile reflects a bias or whether cooperation across pixels contributes to the prediction positively or negatively, and the area of the shade can be (roughly) considered as its strength. As in other studies, we conducted our experiments on ImageNet dataset (Deng et al., 2009). We sampled fifty images to consist of a set of clean images. We generated adversarial images by applying the $L_\infty$-untargeted Iterative-Fast Gradient Sign Method (I-FGSM) with $\epsilon = 16/255$ to the clean images. For the corrupted images, we used ImageNet-C (Hendrycks & Dietterich, 2019) dataset. We sampled fifty images from the images under level-5 *fog* corruption. We used two CNNs and two Vision Transformers pretrained on ImageNet[2], including ResNet-18 (He et al., 2016), AlexNet (Krizhevsky et al., 2012), Swin-T (Liu et al., 2021) and DeiT-Ti (Touvron et al., 2021). More detail on the settings (e.g., image sampling, approximate computation of interactions) is given in Appendix B.1.

In the following, we focus on ResNet-18 and Swin-T. See Appendix B.4 for more results.

### 5.1 MISCLASSIFICATION OF CLEAN IMAGES

Here, we consider the misclassification of clean images. Figure 2(left) shows the distribution of interactions for each order. As can be seen, ResNet-18 did not present a significant difference in the distributions between the images misclassified and those successfully classified. This result indicates that the misclassification is not triggered by the inability of the deep learning models to obtain beneficial cooperation, suggesting that deep learning models gain similar cooperation regardless of the correctness of the classification. As we will show shortly, this is not the case for adversarial and corrupted images. In contrast to ResNet-18, Swin-T has a different tendency for low-order interactions (particularly, at $0.0n$): the interactions from correctly classified images are significantly biased toward positive values, whereas for the misclassified images, these are biased toward negative values. This indicates that for Swin-T, the successfully classified images have local cooperation between pixels that increase the confidence score, whereas the misclassified images do not have such cooperation, leading to the deterioration in the confidence score.

### 5.2 MISCLASSIFICATION OF ADVERSARIAL IMAGES

Next, we consider the misclassification of adversarial images. For each model, the adversarial images are generated by applying I-FGSM to clean images that are correctly classified. Figure 2(center) shows the distribution of interactions for misclassification of adversarial images. ResNet-18 has strongly negative interactions of high order around $0.8n$–$1.0n$. This result indicates that adversarial perturbations in ResNet-18 create spurious cooperation for the model by global shapes and textures. Namely, misclassification by adversarial perturbations is caused by the destruction of the model from the meaningful cooperation between pixels to the spurious one, which is useless or harmful to make a prediction. However, unlike ResNet-18, Swin-T only shows a slight shift of the interactions to negative values, and the large interquartile range covers positive and negative sides almost evenly from middle to high orders. This may explain the empirical robustness of Swin-T compared to ResNet-18 (cf. Table 1 in Appendix A). It is worth noting that this difference between ResNet-18 and Swin-T was observed because we take the sign into account, unlike most of the existing studies. Refer to Appendix B.2 for another example.

### 5.3 MISCLASSIFICATION OF CORRUPTED IMAGES

We now consider corrupted images. Yin et al. (2019) discovered that adversarially trained models are less robust than normally trained ones under the perturbations in a low-frequency domain (e.g.,

---

[1]Although the interaction is defined as Eq. (3), in this paper, we also refer to $\Delta f(i, j, S)$ as interaction for convenience when we are not interested in specific $(i, j, S)$.

[2]https://doi.org/10.5281/zenodo.4431043 (CNNs),
https://github.com/rwightman/pytorch-image-models (Vision Transformers)

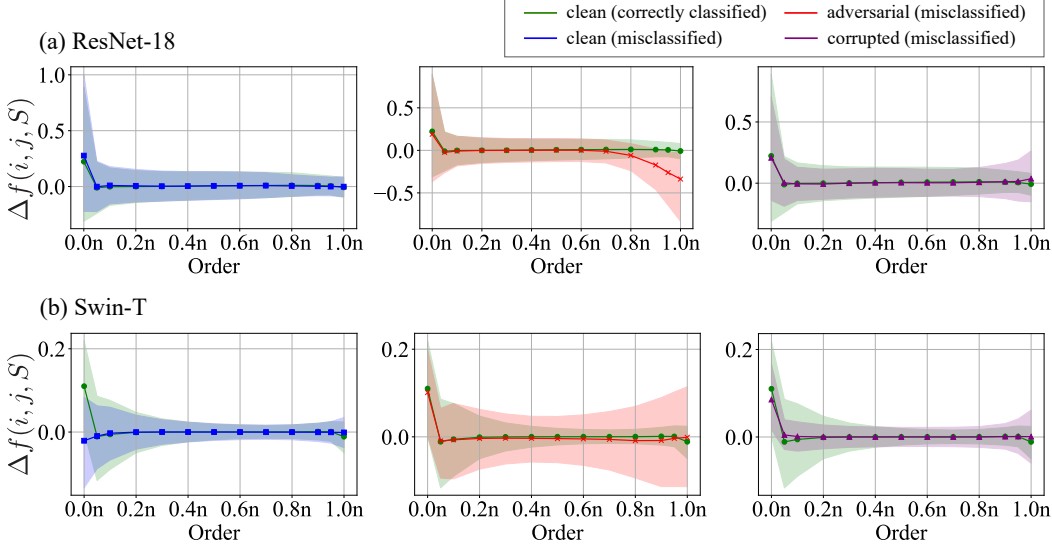

Figure 2: The distribution of interactions (i.e., the distribution of $\Delta f(i, j, S)$ for various pixel pairs $(i.j)$, context $S$, and images). The horizontal axis represents the order of interactions ($n$ denotes the number of pixels in an image). The solid lines represent the median of the interactions. The shades represent the interquartile range. The interactions for misclassified clean (left, blue), adversarial (center, red), and corrupted (right, purple) images are contrasted to those for correctly classified images (green), respectively. One can observe that the three types of misclassified images give different tendencies in the distributions of interactions for each order. The tendency also differs between (a) ResNet-18 and (b) Swin-T.

*fog* corruption). Figure 2(right) visualizes the distribution of interactions for misclassification of the fog-corrupted images. The strength of the interactions for adversarial images is lower at low order and higher at high order than in the case of clean images. This result shows that the corrupted images destroy the local cooperation of pixels that deep learning models were originally able to exploit, and instead, they create spurious global cooperation that is not useful for prediction. Note that compared to the case of adversarial images (Fig. 2; center), the strength of interactions for corrupted images is moderate, particularly in high order, which suggests that as widely known, corruption is less harmful than adversarial perturbations. Swin-T showed a similar but clearer contrast between the distributions of the two image sets than ResNet. In particular, the interactions have a large difference at low orders $0.05n$–$0.1n$. Swin-T shows that local cooperation of fog images is difficult to obtain.

## 5.4 COMPARISON OF MODELS

In Sections 5.1–5.3, Swin-T exhibits different distributions of interactions from ResNet-18. This indicates that Swin-T (Vision Transformer) exploits the different image features from those used by ResNet-18 (CNN). To further investigate this, we compare Swin-T and ResNet-18 by using the distribution of interactions in the clean images that are classified successfully. The result is shown in Fig. 3. ResNet-18 and Swin-T do not show a significant difference in terms of the median; however, at order $0.0n$, the interquartile range includes negative and positive interactions for ResNet-18 but for Swin-T, it only includes positive ones. Thus, the obtained features from cooperation are different for each model, and Swin-T takes advantage more of the local cooperation of pixels than ResNet-18 does. These results may partially explain the higher classification accuracy of Swin-T.

It is worth noting that in Sections 5.1–5.3, one can see that for Swin-T, the distributions of interactions between correctly classified and misclassified images have a clearer difference than those for ResNet-18. This suggests that the characteristics of the predictions can be more clearly exhibited in the case of the Vision Transformer than CNNs, which encourages the further study of Vision Transformers based on interactions.

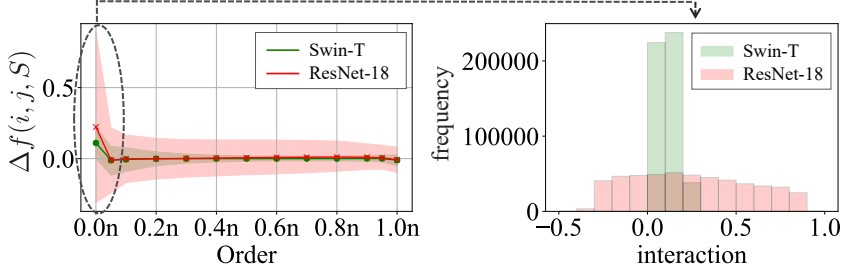

Figure 3: (Left) The distribution of interactions. The horizontal axis represents the order of interactions ($n$ denotes the number of pixels in an image). The two solid lines respectively represent the median of the interactions for clean images that Swin-T (green) and ResNet-18 (red) correctly classified. The shades represent the interquartile range. (Right) at order $0.0n$, the interquartile range for ResNet-18 includes negative and positive interactions; for Swin-T, it only includes positive ones.

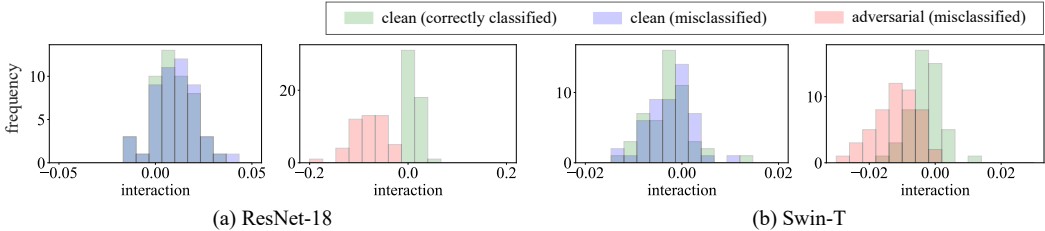

Figure 4: The distribution of average interactions (i.e., the distribution of $\mathbb{E}_{i,j}[I(i,j)]$ for all the pixel pairs $(i,j)$). The distributions for misclassified clean (blue) and adversarial (red) images are contrasted to those for correctly classified clean images (green). The distribution for misclassified clean images is similar to that for correctly classified images, while that for misclassified adversarial images shows a negative transition. This tendency is observed for both (a) ResNet-18 and (b) Swin-T, while the transition is moderate in the latter case.

## 6 ADDITIONAL EXPERIMENTS ON ADVERSARIAL ATTACKS

In Section 5, we observed that adversarial perturbations have significant effects on the distributions of interactions in middle and high order. In this section, we further analyze adversarial images and their transferability by using the interactions. To this end, we consider for each image $\mathbb{E}_{i,j}[I(i,j)]$ as the average of the interaction $I(i,j)$ over all the possible pixel pairs in the image. For the adversarial attacks, we used the same methods and parameters as those used in Section 5.2. Other setups (e.g., dataset and image sampling) are also the same as those in Section 5.

**Negative transition by adversarial attack.** Here, we compare the distribution of average interactions between correctly classified clean images and misclassified (clean or adversarial) images. Figure 4 shows the histograms of the average interactions of images in the dataset. As demonstrated in Fig. 4(a), where ResNet-18 is considered, the distribution of average interactions is similar between the two clean image sets (correctly classified clean images and misclassified ones), whereas there is a large shift to negative values for adversarial images. The same observation holds for Swin-T as shown in Fig. 4(b). The negative shift of average interactions for adversarial images means that the pixel cooperations that models use to increase the confidence score are damaged or lost. Note that the negative shift is more moderate for Swin-T than ResNet-18. This indicates that although the images are misclassified, the damage in the cooperation of pixels is moderate for Swin-T. We consider that this reflects the fact that Swin-T is more robust than ResNet-18 (cf. Table 1 in Appendix A).

**Transferability and interactions** We investigate adversarial transferability from the perspective of multi-order interactions. We consider $\mathbb{E}_{(i,j)}\left[I^{(s)}(i,j)\right]$, which is the average of interactions of

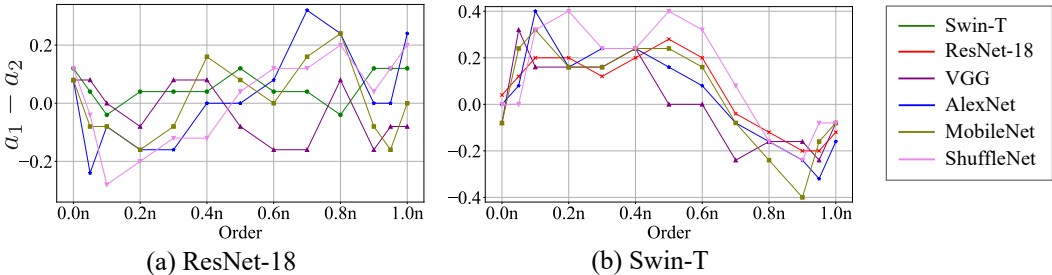

(a) ResNet-18          (b) Swin-T

Figure 5: The difference in attack success rates between the sets of images with high and low average interactions (i.e., $a_1 - a_2$). The horizontal axis represents the order of interactions ($n$ denotes the number of pixels in an image). (a) For ResNet-18, adversarial images with higher interactions in high order transfer better, whereas those with lower interactions transfer less. (b) For Swin-T, the trend is opposite to that for ResNet-18.

order $s$ over all the pixel pairs. As in Section 5.2, we generate two sets of adversarial images on ImageNet using ResNet-18 and Swin-T, respectively. We first focus on ResNet-18. We divided the set of the adversarial images into two sets, $D_1$ and $D_2$, by the average interactions. The first set $D_1$ contains the adversarial images whose average interactions are higher than the median, while the second set $D_2$ contains the rest. We then measure the transferability of adversarial images in each set $D_i$ by the attack success rate $a_i$, which is defined as the ratio of the number of successfully attacked images in a source model to that in a target model. We used ResNet-18 and Swin-T as source models and ResNet-18, Swin-T, AlexNet, ShuffleNet (Ma et al., 2018), and MobileNet (Sandler et al., 2018) as target models. These models were pretrained on ImageNet[3]. In Fig. 5, the difference in attack success rates, $a_1 - a_2$, between the sets of images with high and low average interactions is plotted along the orders. As shown in Fig 5(a), for $D_1$ (i.e., adversarial images with high average interactions), the transferability increases as the order increases. Interestingly, as shown in Fig 5(b), for $D_2$ (i.e., adversarial images with low average interactions), the trend is the opposite: the traceability *decreases* along the order. To summarize, adversarial images with higher interactions in high order transfer better when these are generated by using ResNet-18 but transfer less when Swin-T is used, and vice versa. This contrastive tendency is similar to the recent observation that the adversarial images generated by CNNs are more perturbed in the high-frequency domain, whereas those generated by Vision Transforms are more perturbed in the low-frequency domain (Kim & Lee, 2022).

# 7   CONCLUSION

This study conducted the first analysis of the misclassification of various types of images from a game-theoretic perspective. We considered clean, adversarial, and corrupted images and characterized each of them using the distribution, order, and sign of interactions. Our extensive experiments revealed that each type of misclassification has a different tendency in the distribution of interactions, suggesting that the model relies on different categories of image features to make an incorrect prediction for each image type. We also analyzed Vision Transformers using interactions for the first time, finding that they give a distribution of interactions that is different from that of CNNs in some cases, which confirms the recent observations that Vision Transforms exploit more robust features than CNNs do from a new perspective. Our experiments also provide a new result that contrasts Vision Transformers to CNNs: CNNs generate more transferable adversarial images from those with high interactions in high order, but the trend is the opposite for Vision Transformers. This paper reported various observations on CNNs and Vision Transformers based on interactions. We believe that this will contribute to deepening the understanding of machine vision.

---

[3] https://doi.org/10.5281/zenodo.4431043 (CNNs),
https://github.com/rwightman/pytorch-image-models (Vision Transformers)

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

## A    ADVERSARIAL ROBUSTNESS EVALUATION

We evaluated the adversarial robustness of ResNet-18 and Swin-T by the success rate of the $L_\infty$-untargeted I-FGSM attack with various perturbation strength ($\epsilon = 0.5/255, 1/255, 2/255, \cdots$). For each model, we selected 1,000 clean images that are classified correctly. The result is shown in Table 1. Swin-T has a lower attack success rate than ResNet-18 for all $\epsilon$, showing that the former is more adversarially robust than the latter.

## B    EXPERIMENTAL SETUP AND ADDITIONAL EXPERIMENTS

### B.1    EXPERIMENT DETAILS

As the computation of interactions is known as an NP-Hard problem, we reduce the computational cost as follows. We divided each image into $16 \times 16$ patches and measured the interactions between the patches. Instead of using the full dataset, we randomly sampled fifty images of different classes. For each image, 200 patch pairs are randomly selected, and for each

Table 1: The success rate of the I-FGSM attack with various perturbation strengths. Swin-T shows a lower attack success rate than ResNet-18.

| $\epsilon$ | ResNet-18 | Swin-T |
|------|-----------|--------|
| 0.25 | 0.652 | 0.545 |
| 0.5 | 0.924 | 0.767 |
| 1 | 0.996 | 0.937 |
| 2 | 1.0 | 0.993 |
| 4 | 1.0 | 1.0 |
| 16 | 1.0 | 1.0 |

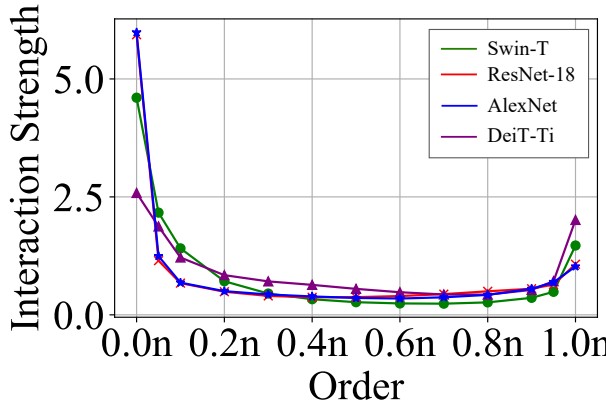

Figure 6: The median of interaction strength Eq. (B.2) along the order. The solid lines represent the median. The trend is similar across models. Compare to Figs. 2(left) and 8(left).

patch pair, 100 contexts are randomly selected. Here, each patch pair is randomly sampled so that one patch is within a radius of two patches from the other patch. For order, we consider $s = 0.0n, 0.05n, 0.1n, 0.2n, \cdots, 0.8n, 0.9n, 0.95n, 1.0n$. The abovementioned setup is the same as in (Deng et al., 2022). In addition, when we compute an interaction $I(i,j)$, we average $I^{(s)}(i,j)$ for $s = 0.0n, 0.1n, \cdots, 1.0n$.

## B.2 STRENGTH OF INTERACTIONS

Most of the existing studies used the strength of an interaction, which is computed as follows (Cheng et al., 2021; Deng et al., 2022).

$$J^{(m)} = \frac{\mathbb{E}_{x \subset \Omega}\left[\mathbb{E}_{i,j}\left[\left|I^{(m)}(i,j)\right|\right]\right]}{\mathbb{E}_{m'}\left[\mathbb{E}_{x \subset \Omega}\left[\mathbb{E}_{i,j}\left[\left|I^{(m')}(i,j)\right|\right]\right]\right]}$$

By contrast, our study considers the sign of interactions. As we discussed in 4.3, the sign tells us whether the pixel pair interacts positively or not, and there are several trends that cannot be grasped if one only considers the strength of interactions. Fig. 6 shows an example, where we plotted the median of interactions for clean images along the order. As can be seen, for all the models, the trend of interaction strength is similar (i.e., high interaction strength at low and high orders). However, as shown in Figs. 2(left) and 8(left), the four models present different trends when singed interactions are used.

## B.3 CORRUPTED IMAGES (*Gaussian noise*)

In Section 5.3, we used *fog* corruption because this is known as the low-frequency corruption, in which adversarially trained models are less robust than normally trained ones. Here, we conduct the same experiments with Gaussian noise corruption, for which adversarially trained models are known to perform better than normally trained ones. The result is shown in Fig. 7. One can see that

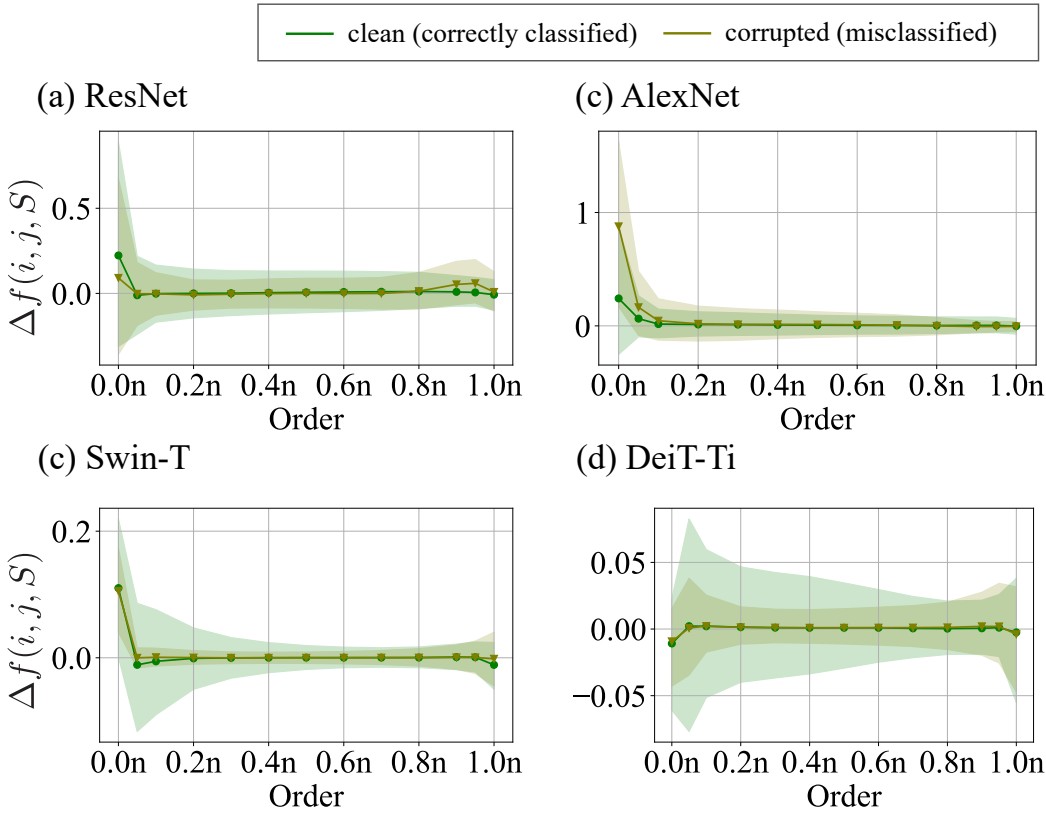

Figure 7: The distribution of interactions for misclassified images under *Gaussian noise*. Compare to Figs. 2(right) and 8(right).

for all the models except AlexNet, the strength of low-order interactions is low, while the strength of high-order interactions is high. This trend is similar to that for *fog* corruption (cf. Figs. 2(right) and 8(right)); thus, the misclassifications under *fog* and *Gaussian noise* can be considered similar in a sense.

### B.4 RESULTS FOR ALEXNNET AND DEIT-TI

We investigated whether our experiments in Section 5 generalize to other models. We used AlexNet and DeiT-Ti that are pretrained on ImageNet. The experimental setup is the same as in Section 5 The result is shown in Fig. 8. The tendency of interactions for AlexNet was the same as that for ResNet-18. DeiT-Ti also presented almost the same tendency as Swin-T. However, the tendency in the low-order interactions for clean images was different. Swin-T showed a striking difference in the sign of the interactions between correctly classified and misclassified images; Deit-Ti showed no distinct difference. This result indicates that the importance of low-order interactions for the prediction differs between models.

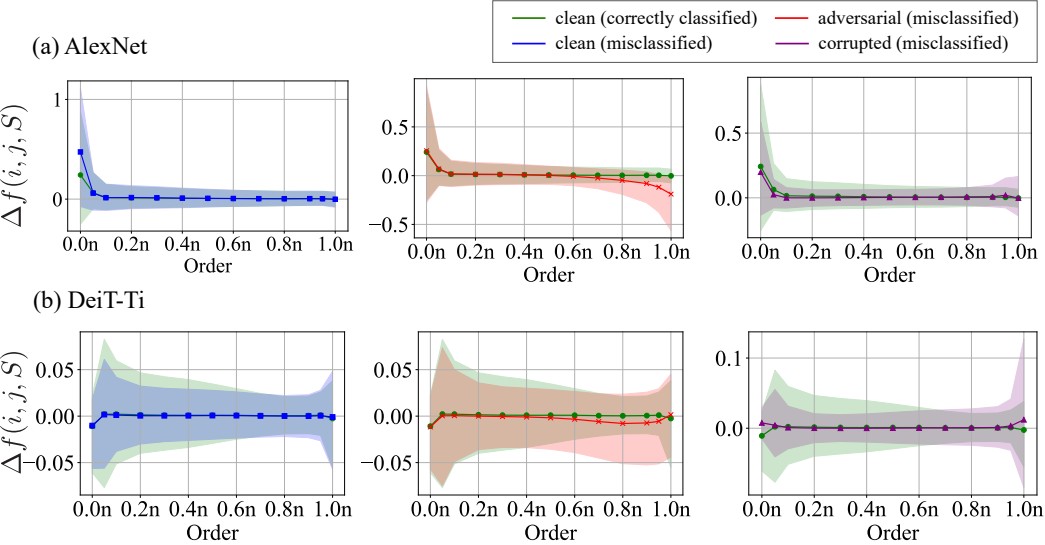

Figure 8: The distribution of interactions (i.e., the distribution of $\Delta f(i, j, S)$ for various pixel pairs $(i.j)$, context $S$, and images). The horizontal axis represents the order of interactions ($n$ denotes the number of pixels in an image). The solid lines represent the median of the interactions. The shades represent the interquartile range. The interactions for misclassified clean (left, blue), adversarial (center, red), and corrupted (right, purple) images are contrasted to those for correctly classified images (green), respectively. One can observe that the three types of misclassified images give different tendencies in the distributions of interactions for each order. Compare to Fig. 2.

