# OpenReview forum: "Game-Theoretic Understanding of Misclassification"
_ICLR.cc/2023/Conference — Submitted to ICLR 2023_

### Official Review · Reviewer_QLY7 · 2022-10-22

**Confidence:** 4
**Correctness:** 3
**Technical Novelty And Significance:** 2
**Empirical Novelty And Significance:** 3
**Recommendation:** 3

**Clarity, Quality, Novelty And Reproducibility:**

The paper has several clarity issues and although the investigation of misclassification with interactions is novel, comparison to other diagnostic approaches is necessary.

**Strength And Weaknesses:**

### Things I liked

The paper considers an interesting angle to investigate misclassification of a classifier and is able to highlight interesting pattens from the study (eg. corrupted and adversarial images impact different kind of interactions and order).

### Things that need clarification / improvement

- The authors highlight that Cheng et. al. 2021 argue that lower-order reflects local properties while higher order reflects global properties (Sec 4.2). Unfortunately, I am not entirely convinced. I would rather only consider the 1.0n case which measures the interaction of a pair of pixels in the context of the entire input image.

- The paper is incomplete in many ways.
  - While the authors do an analysis of the multiple input image classes on which a classifier misclassifies, I am not sure how one can use it to improve a classifier robustness. For example, does the interesting observation that corrupt and adversarial images impact different types/orders of interaction have any conclusion like existing defense mechanisms cannot target both ends and fixing on will not help fix the other?
  - How does data augmentation or other defense strategies affect these interactions?
  - Why is interaction-based diagnosis better/more useful compared to other debugging approaches (eg. GradCAM, analyzing the impact of individual pixels on loss that is used to craft adversarial examples).

- The authors simply use a shapely value type measure to characterize the contribution of pixel pairs to prediction via masking. This is not strong enough a motivation to say they have done a game-theoretic understanding of misclassification, as the title suggests.

- There are several clarity issues with the paper.
  - In Figure 1 (a), one can form many pairs of black pixels. Do all combination of these pairs have $\delta f(i,j,S)$ as strongly positive?

- In Figure 2, why is the curve for correctly classified clean examples (in green) different in the middle figure for ResNet-18 (i.e. top row) compared to the left and right one? I was expecting the clean curves to be similar in all the 3 graphs for a row (which also seems to be the case for Swin-T).

- In Figure 3 and Figure 4, should the distribution of clean images not be the same in the frequency charts. For example, I am a bit confused that Swin-T has only green bars in the positive side for the clean images in Fig 3 (left) but has a normal looking distribution for clean images in Fig 4 (b). What is the n for the distribution in Fig 4, I see it is an expected value over I(i,j) but I wasn't sure where the difference is coming form?

Other corrections:

- [Sec 5.3] "The strength of the interactions for adversarial images is lower" -> "The strength of the interactions for corrupted images is lower"


**Summary Of The Paper:**

The paper diagnosis the behavior of CNN and Transformer models on three classes of images-- clean, adversarial, and real-world corruptions-- for the ImageNET dataset. For this purpose, they use `interactions'-- shapely values that determine how much a pair of pixels contribute to the class prediction, given a set of pixels in the image (that determine the order of interaction). Additionally, the authors consider the sign of interaction which helps them determining if the pixels together help increase/decrease the confidence of prediction.

**Summary Of The Review:**

The paper, although interesting in how it tries to diagnose misclassification in CNN and transformer models for image domain via interactions, is incomplete in several regards. It does not compare it self to any other diagnostic took, it does not highlight the usefulness of the diagnostic tool.

---

> ### Author Response · Authors · 2022-11-15
> **Response to Reviewer QLY7, Part 2**
>
>
> > In Figure 3 and Figure 4, should the distribution of clean images not be the same in the frequency charts. For example, I am a bit confused that Swin-T has only green bars in the positive side for the clean images in Fig 3 (left) but has a normal looking distribution for clean images in Fig 4 (b). What is the n for the distribution in Fig 4, I see it is an expected value over I(i,j) but I wasn't sure where the difference is coming form?
>
> Figure 3 and Figure 4 show different distributions of clean images. Figure 3(left) presents the distribution of $\Delta f(i,j,S)$, and Figure 3(right) shows its histogram at order $s=0.0n$. However, Figure 4(b) presents a histogram of $\mathbb{E}_{(i,j)}(I(i,j))$, where $I(i,j) = \sum_S \Delta f(i,j,S)$. Interaction $I(i, j)$ can be considered as the average of multi-order interactions $I^{(s)}(i,j)$ over order $s$ (see the last line of Section 3), and thus no order (e.g. "$s=0.0n$") is defined for this quantity.
>
> - [Wang et al., 2021] Xin Wang, Jie Ren, Shuyun Lin, Xiangming Zhu, Yisen Wang, Quanshi Zhang： A Unified Approach to Interpreting and Boosting Adversarial Transferability. ICLR 2021
> - [Zhang et al., 2021] Hao Zhang, Sen Li, YinChao Ma, Mingjie Li, Yichen Xie, Quanshi Zhang： Interpreting and Boosting Dropout from a Game-Theoretic View. ICLR 2021
> - [Cheng et al.,2021] Xu Cheng, Chuntung Chu, Yi Zheng, Jie Ren, Quanshi Zhang： A Game-Theoretic Taxonomy of Visual Concepts in DNNs. arXiv preprint 2021
> - [Ren et al.,2021] Jie Ren, Die Zhang, Yisen Wang, Lu Chen, Zhanpeng Zhou, Yiting Chen, Xu Cheng, Xin Wang, Meng Zhou, Jie Shi, Quanshi Zhang： Towards a Unified Game-Theoretic View of Adversarial Perturbations and Robustness. NeurIPS 2021

---

> ### Author Response · Authors · 2022-11-15
> **Response to Reviewer QLY7, Part 1**
>
> We thank the reviewer for the thoughtful suggestions and detailed reviews.
>
> > While the authors do an analysis of the multiple input image classes on which a classifier misclassifies, I am not sure how one can use it to improve a classifier robustness. For example, does the interesting observation that corrupt and adversarial images impact different types/orders of interaction have any conclusion like existing defense mechanisms cannot target both ends and fixing on will not help fix the other?
>
> > How does data augmentation or other defense strategies affect these interactions?
>
> Our study obtained various interaction-based observations from a novel perspective (i.e., misclassification) to build robust models. One of the examples is, as you put as an example, corrupted and adversarial images have different distributional shifts in interactions. This observation suggests that these two types of perturbed images fool the classifier for different causes, supporting the empirically known fact that attaining robustness for these two types of perturbations is difficult from a new perspective. Several studies proposed methods to introduce interactions in the loss function, so with their methods and our observations, we can design robust models or strong attacks. This is an interacting direction to improve our study.
>
> > Why is interaction-based diagnosis better/more useful compared to other debugging approaches (eg. GradCAM, analyzing the impact of individual pixels on loss that is used to craft adversarial examples).
>
> Interaction-based diagnosis, which focuses on cooperation between pixels, provides new insights that cannot be obtained by other approaches that focus on individual pixels (e.g., GradCAM). For example, [Wang et al., 2021] found that the interactions can serve as a strong (negatively-correlated) indicator of adversarial transferability of adversarial examples and proposed the first method to generate adversarial examples while controlling their transferability.
>
> Interaction-based diagnosis in computer vision is a recently emerging topic only starting in 2021 by [Zhang et al., 2021], and what objects and how they can be analyzed using interaction-based diagnosis are still not fully explored. Our study examines various types of image misclassification using interaction-based diagnosis, which has not been targeted before. We found that the distribution of multi-order interactions varies across the types of misclassification. We also provide the first analysis of Vision Transformers using interactions. We found that Vision Transformers show a different tendency in the distribution of interactions from that in CNNs.
>
> In addition to that perspective, this paper proposes to measure the distribution, sign, and order of the interactions so that one can capture more fine-grained difference between targets (i.e., type of misclassification and architectures). With this new measure, we obtained new insights into the different types of misclassification and architectures. Many of these insights cannot be obtained from the analysis using interaction intensity in existing studies.
>
> > The authors simply use a shapely value type measure to characterize the contribution of pixel pairs to prediction via masking. This is not strong enough a motivation to say they have done a game-theoretic understanding of misclassification, as the title suggests.
>
> Our title follows the convention. [Ren et al.,2021; Cheng et al.,2021; Zhang et al.,2021] performed analysis using the interactions, and all of them are titled "game-theoretic". Our study is in the same line as these studies.
>
> > In Figure 1 (a), one can form many pairs of black pixels. Do all combination of these pairs have $\Delta f(i,j,S)$ as strongly positive?
>
> No, Figure 1(a) shows many pixel pairs with highly positive $\Delta f(i,j,S)$ simultaneously; namely, each pixel in black has a specific partner. We will consider a better presentation, e.g., coloring each pair differently.
>
> > In Figure 2, why is the curve for correctly classified clean examples (in green) different in the middle figure for ResNet-18 (i.e. top row) compared to the left and right one? I was expecting the clean curves to be similar in all the 3 graphs for a row (which also seems to be the case for Swin-T).
>
> In Figure 2 in ResNet-18, the curves for correctly classified clean examples are all the same in the left, middle, and right figures. They may appear different only because the scale of the vertical axes is different across the figures. We consider that this is favorable to present better the difference of the distribution of interactions between clean and adversarial images.

---

### Official Review · Reviewer_zrJN · 2022-10-24

**Confidence:** 3
**Clarity, Quality, Novelty And Reproducibility:** Mostly clearly written, seems reprodu…
**Correctness:** 4
**Technical Novelty And Significance:** 2
**Empirical Novelty And Significance:** 2
**Recommendation:** 5

**Strength And Weaknesses:**

Studying when and why deep models misclassify is an important problem. The paper gives some insights on this problem through the lens of interaction between pixels. They study misclassification of clean images and the ones with adversarial and non-adversarial corruptions. For the clean images they observe that the successfully classified images have local cooperation between pixels that increase the confidence score, while the misclassified images do not have such cooperation, leading to the deterioration in the confidence score. For the adversarially corrupted images they observed that misclassification by adversarial perturbations is caused by the destruction of the model from the meaningful cooperation between pixels to the spurious one, which is useless or harmful to make a prediction, while these harmful effects are milder for non-adversarially corrupted images. Further, they conducted same study with Vision transformers and found that Vision Transformer more clearly exhibit the prediction characteristics than CNNs.

Weaknesses/Questions,
Could you provide more intuitive understanding of interaction ( something like Figure 1). Why are the left and middle figures roughly the same in Figure 1 but have opposite signs of interaction?
 There are a ton of methods to generate adversarial images, will the paper's observations remain applicable to these diverse set of adversarial methods? Soon one can design a method to generate adversarial images that show no difference in terms of interaction as well. Or can this be shown that it is not possible?
Since the conclusions are drawn from experiments alone, it would be more helpful to see if the similar observations hold for various datasets and models as well?

**Summary Of The Paper:**


The paper studies various types of misclassification of images by deep neural networks. The study is done using a quantity called  "interaction" between pixels. This quantity is motivated from game theoretic concept of quantifying interaction between players in a co-operative game. This quantity has been used in other studies on deep learning. ( Cheng et al., 2021; Deng et al., 2022; Ren et al., 2021 and others).  The paper characterizes the misclassification of clean, adversarial, and corrupted images with the distribution, order, and sign of the interactions. They find that each type of misclassification has different tendencies in interactions, which indicates that each type of misclassification is triggered by different causes. They also provide an analysis of Vision Transformers by using interactions and report that the difference in distributions of interactions between misclassified and correctly classified images are clearer and also different from the case with CNNs. Further they also found that the images that are more adversarially transferable have the opposite tendency in the interactions between Vision Transformers and CNNs.


**Summary Of The Review:**

The paper studies and important problem of understanding misclassification by neural nets and studying it through the lens of interaction is novel. Their empirical observations give insights into various types of misclassifications. I have a few questions/concerns listed above that could improve my assessment of the paper.

---

> ### Author Response · Authors · 2022-11-15
> **Response to Reviewer zrJN**
>
> We thank the reviewer for the careful reading and insightful questions.
>
> > Could you provide more intuitive understanding of interaction ( something like Figure 1). Why are the left and middle figures roughly the same in Figure 1 but have opposite signs of interaction?
>
> The interaction measures the cooperative contribution of image pixels to the prediction, such as an edge in an image. Recent studies have revealed that the interactions capture not only the edge features but also shape and texture features. For example, [Cheng et al., 2021] showed that low-order interactions reflect local shapes and textures, whereas high-order interactions reflect global shapes and textures. The positive (negative) interactions indicate that two pixels cooperatively increase (decrease) the confidence score given to the correct class.
>
> Figures 1(left) and 1(right) visualize pixel pairs with highly positive and negative interactions, respectively. In both cases, the pixel pairs cluster around the shark's body. We consider that this is because the model pre-trained on ImageNet learns that objects in images should strongly influence the classification. Thus, the magnitude of interactions between pixel pairs tends to be large around objects (i.e., shark). Ideally, all the pixel pairs around the shark should have positive interactions, but as shown in Figure 1(middle), this is not the case here. We suspect that this results from the model's overfitting to non-generalizable features and/or underfitting to generalizable features in the training samples. As even well-trained models misclassify several images due to their overfitting and underfitting to the training samples, models can give negative interactions to certain pixel pairs (a sort of "misclassification" at local level). Because the computation of an interaction (more specifically, here, $\Delta f(i, j, S)$) is done for a masked image, where only $i$-th and $j$-th pixels and context $S$ are visible, such local-level misclassification can happen.
>
> > There are a ton of methods to generate adversarial images, will the paper's observations remain applicable to these diverse set of adversarial methods?
>
> We consider that the observations remain applicable to a wide range of other adversarial attacks (except those under special restrictions) because (i) the common goal of them is to change the confidence score and (ii) it has been known that the adversarial training with a specific attack (e.g., Projected Gradient Descent; PGD) can yield a model that is robust for various attacks to some extent. In our paper, we used I-FGSM as the attack method, but after reading your comments, we conducted the same experiment with PGD and obtained similar results.
>
> > Soon one can design a method to generate adversarial images that show no difference in terms of interaction as well. Or can this be shown that it is not possible?
>
> We appreciate your interesting question. Yes, as numerous studies have shown that one can generate adversarial examples under various constraints even when the constraints seem very restrictive, we consider that designing such interaction-invariant attack is possible, and this can be new work. Namely, it is interesting to investigate its efficient implementation (as interaction computation is costly), the qualitative difference from other adversarial examples, and the defense method against the attack. In Figure 5, we show that the transferability of an adversarial image can be estimated from the interaction of its (non-perturbed) original image. The interaction-invariant attack may avoid the detection of adversarial examples based on this observation, and thus, it is worth investigating such an attack.
>
> > Since the conclusions are drawn from experiments alone, it would be more helpful to see if the similar observations hold for various datasets and models as well?
>
> We will conduct more experiments using other datasets and models, but we also consider that the current experimental setting well supports our observations and conclusions. As for the models, we considered four widely used models (ResNet-18, AlexNet, Swin-T, DeiT-Ti). As for the dataset, we used ImageNet, which is commonly used in related studies on interactions. Some of these studies consider one more dataset, but it varies across them. Thus, we decided to focus on ImageNet.
>
> - [Cheng et al., 2021] Xu Cheng, Chuntung Chu, Yi Zheng, Jie Ren, Quanshi Zhang： A Game-Theoretic Taxonomy of Visual Concepts in DNNs. arXiv preprint 2021

---

### Official Review · Reviewer_oeXE · 2022-10-25

**Confidence:** 1
**Clarity, Quality, Novelty And Reproducibility:** In general, I think this paper is wel…
**Correctness:** 3
**Technical Novelty And Significance:** 3
**Empirical Novelty And Significance:** 3
**Recommendation:** 6

**Strength And Weaknesses:**

Strength:
This work conducted extensive experiments to support the idea. The Shapley value interaction seems to be an interesting idea.
Weakness:
1. The difference between distributions of interactions for different types of images is quite small in Figure 2. Especially between clean and corrupted images and low order of interactions. I was wondering if the authors can provide some explanations for this.
2. More images like the visualization in Figure 1 may need to be provided in the appendix to make the intuition more convicing.

**Summary Of The Paper:**

This paper investigates various types of misclassifications from a game-theoretic perspective. The authors study the misclassification of clean, adversarial, and corrupted images, and found that the dominant order of interactions is different between the three types of misclassification. The authors also found that the interactions of ViT are also different from CNNs.

**Summary Of The Review:**

I think the general idea of this paper is interesting. The authors provided some interesting perspectives for adversarial examples through Shapley value interactions.

---

> ### Author Response · Authors · 2022-11-15
> **Response to Reviewer oeXE**
>
> We thank the reviewer for the careful reading and the suggestions for improvement.
>
> > The difference between distributions of interactions for different types of images is quite small in Figure 2. Especially between clean and corrupted images and low order of interactions. I was wondering if the authors can provide some explanations for this.
>
> We consider that the difference between distributions across types of misclassification is large in general; each type of image has its own characteristics in the distribution. Even between clean and corrupted images, which you are concerned about, one can see the latter has a moderate variance from middle to high order, whereas the former does not.
> At low order, we agree that the difference is small among the three types of images but only in the ResNet-18 case. Note that at low order, or particularly at $s=0.0n$, only two pixels of interest are visible and other parts are masked. Thus, it is very hard for models to capture stable image features. We consider that this is the reason for the large variance at low order and the similarity across the image types. This should be applied to Swin-T, but we observed that this is not the case for Swin-T. As many recent studies suggested, CNNs and Vision Transformers exploit different features (e.g., Vision Transformers exploit more shape information [Tuli et al., 2021]). We suspect that our observation also results from the fundamental difference in architecture between CNNs and Vision Transformers, but further analysis is needed to conclude so.
>
> > More images like the visualization in Figure 1 may need to be provided in the appendix to make the intuition more convincing.
>
> We will add more examples as Figure 1 to the Appendix.
>
> - [Tuli et al., 2021] Shikhar Tuli, Ishita Dasgupta, Erin Grant, Thomas L. Griffiths： Are Convolutional Neural Networks or Transformers more like human vision? arXiv preprint 2021

---

### Official Review · Reviewer_Mq4K · 2022-10-25

**Confidence:** 3
**Correctness:** 3
**Technical Novelty And Significance:** 2
**Empirical Novelty And Significance:** 3
**Recommendation:** 6

**Clarity, Quality, Novelty And Reproducibility:**

This paper is well-written and understanding misclassification via interactions among pixels is interesting.

**Strength And Weaknesses:**

Originality:

1. Focusing misclassifications by using interactions among pixels is interesting, which manifests three types of misclassifications have a distinct tendency in interactions and each of them arises from different causes.

2. It also analyzes Vision Transformers by using interactions to show their feature extraction that is different from CNNs.

Question:

Adversarial training is a well-known standard method to improve models' robustness. However, misclassification seems to be inevitable even if model is trained by adversarial training. So, dose there exist a significant difference between the interactions of misclassifications original models and those trained by adversarial training?

**Summary Of The Paper:**

This work empirically investigates various types of image misclassification from a game-theoretic view, which includes clean images, adversarially perturbed images and corrupted images. It characterizes the misclassification with the distribution, order, and sign of the interactions, and numerical results show three types of misclassifications have different tendencies in interactions. And it also explores interactions of misclassifications in different model architectures, CNNs and Vision Transformers, which implies Vision Transformers may exploit the features that CNNs do not use for the prediction.

**Summary Of The Review:**

This work studies misclassification of various types of images from a game-theoretic perspective. It provides a novel understanding of misclassification for different types of images and model architectures.

---

> ### Author Response · Authors · 2022-11-15
> **Response to Reviewer Mq4K**
>
> We would like to thank you for your careful reading of our paper, comments, and an interesting suggestion.
>
> > Adversarial training is a well-known standard method to improve models' robustness. However, misclassification seems to be inevitable even if model is trained by adversarial training. So, dose there exist a significant difference between the interactions of misclassifications original models and those trained by adversarial training?}
>
> Although thorough experiments are required, there should exist a difference in the interactions between original models (normally trained models) and those trained by adversarial training.
> We consider so because numerous studies show that adversarial training yields models that have many different properties from various aspects. To list a few,
>
> - [Santurkar et al., 2019] attacked the images by a strong targeted PGD attack with adversarially trained models and showed that the attacked images have the feature of targeted classes in a human-aligned way. This indicates that adversarially trained models exploit the image features that align with human perceptions more than normally trained models.
>
> - [Yin et al., 2019] revealed that adversarially trained models focus more on low-frequency features of images than normally trained models, and thus, these models are less robust against common corruptions in low frequency than normally trained models.
>
> If not restricted to adversarially trained models,
>
> - [Chen et al., 2021] showed that robust models against common corruptions could be generated by encouraging models to exploit phase features rather than amplitude features as humans do.
>
> It is highly likely that interactions can capture the different natures of non-robust and robust models, and thus, the analysis of robust models through the lens of interactions can be interesting future work.
>
> - [Santurkar et al., 2019] Shibani Santurkar, Dimitris Tsipras, Brandon Tran, Andrew Ilyas, Logan Engstrom, Aleksander Madry: Image Synthesis with a Single (Robust) Classifier. NeurIPS 2019
> - [Yin et al., 2019] Dong Yin, Raphael Gontijo Lopes, Jonathon Shlens, Ekin D. Cubuk, Justin Gilmer: A Fourier Perspective on Model Robustness in Computer Vision. NeurIPS 2019
> - [Chen et al., 2021] Guangyao Chen, Peixi Peng, Li Ma, Jia Li, Lin Du, Yonghong Tian: Amplitude-Phase Recombination: Rethinking Robustness of Convolutional Neural Networks in Frequency Domain. ICCV 2021

---

### Official Review · Reviewer_C5ku · 2022-10-25

**Confidence:** 5
**Correctness:** 3
**Technical Novelty And Significance:** 1
**Empirical Novelty And Significance:** 1
**Recommendation:** 1

**Clarity, Quality, Novelty And Reproducibility:**

Clarity: good;
Reproducibility: good;
Quality: poor;
Novelty: limited;

**Strength And Weaknesses:**

[Weakness]

1. This paper has very limited novelty. In fact, the main content of this paper is very similar to the line of previous studies (Cheng et al., 2021; Deng et al., 2022; Ren et al., 2021; Wang et al., 2021; Zhang et al., 2021), which proposed to use the distribution of interactions encoded in input images to study the feature/conceptual representation of a DNN. This paper directly adopts theories, algorithms, and experimental designs in previous studies, and the core difference is that this paper uses different types of input images and different types of DNNs in experiments. Specifically, it can be found that in Sections 3 and 4, the main contents of introducing (multi-order) interactions are similar to previous work(Cheng et al., 2021; Deng et al., 2022; Ren et al., 2021; Wang et al., 2021; Zhang et al., 2021).

2. Some conclusions also overlap with those of previous studies. For example, Figure 2 shows that adversarial attacks make DNNs use more negative high-order interactions, which has been found in previous work(Ren et al., 2021). Besides, this is the only significant conclusion in Figure 2, while the difference in the first and last columns of Figure 2 is marginal.

3. Experiments in the paper are not sufficient. In the main paper, the authors only conduct experiments on two models on a dataset, and only one setting is used for adversarial attack and corruption, respectively. Therefore, it is questionable whether the obtained conclusions also appear in other DNNs/datasets/settings. I suggest the authors conduct more experiments on more architectures with more settings to demonstrate the validity of their conclusions.

4. The comparison between ResNet-18 and Swin-T in Figure 3 is not convincing. First, the comparison may be unfair. The magnitudes of outputs of the two models may be different, so the value range of $\Delta f(i,j,S)$ may be different in the two models. Therefore, interactions in the two models cannot be directly compared. Second, the 0-order interaction in Figure 3 is significantly affected by baseline values in the computation of interactions, and the authors do not discuss this problem. Third, the difference in the interaction median between the two models is not significant, and it is hard to judge whether the difference is caused by other factors like initialization, dataset, or the training method. The phenomenon on only one pair of models cannot support the conclusion.

5. The authors do not provide any theoretical explanations for conclusions in the paper.

6. The experiment and conclusion about adversarial transferability in Section 6 are not convincing. When comparing the transferability of inputs with different interactions of a specific order, the interactions of other orders are not controlled to be the same. Therefore, the high transferability of inputs with high 0.8n-order interactions may be due to interactions of other orders. This experiment cannot provide a fair comparison between different orders.

In general, in my opinion, this paper neither presents theoretical (algorithmic) contributions, nor introduces new experimental designs so as to provide new insights. More crucially, the conclusions also have large overlap with conclusions of previous studies.

**Summary Of The Paper:**

Similar to previous studies, this paper studies the feature representation of a DNN by empirically investigating the distribution of interactions of input images. On the one hand, the authors investigate interactions of three types of mis-classified input images (including clean, adversarial, and corrupted images). On the other hand, the authors investigate interactions on two types of DNNs, including CNNs and version Transformers. Finally, the authors obtain some conclusions by analyzing tendencies in encoding interactions in these settings.

**Summary Of The Review:**

This paper has very limited novelty. In fact, the main content of this paper is very similar to previous studies (Cheng et al., 2021; Deng et al., 2022; Ren et al., 2021; Wang et al., 2021; Zhang et al., 2021), which proposed to use the distribution of interactions encoded in input images to study the feature/conceptual representation of a DNN. This paper directly adopts theories, algorithms, and experimental designs in previous studies, and the core difference is that this paper uses different types of input images and different types of DNNs in experiments. Specifically, it can be found that in Sections 3 and 4, the main contents of introducing (multi-order) interactions are similar to previous work (Cheng et al., 2021; Deng et al., 2022; Ren et al., 2021; Wang et al., 2021; Zhang et al., 2021).
Moreover, some conclusions also overlap with those of previous studies. For example, Figure 2 shows that adversarial attacks make DNNs use more negative high-order interactions, which has been found in previous work (Ren et al., 2021). Moreover, this is the only significant conclusion in Figure 2, while the difference in the first and last columns of Figure 2 is marginal.

---

> ### Author Response · Authors · 2022-11-15
> **Response to Reviewer C5ku, Part 3**
>
> > The authors do not provide any theoretical explanations for conclusions in the paper.
>
> Currently, our study focuses on collecting empirical observations of interactions in various types of misclassification, and the theoretical analysis of them will be future work. We consider that the theoretical analysis will be a significant challenge because it needs to model clean, corrupted, and adversarial images mathematically (and then, link them to interactions). To the best of our knowledge, there is no good mathematical model to handle clean and corrupted images except extremely restricted ones (e.g., assuming that "clean images" follow a Gaussian distribution). Thus, while we fully agree that theoretical analysis is important, we believe that the collection of our empirical observations will have a sufficient contribution to future studies that use interaction-based analysis.
>
> > The experiment and conclusion about adversarial transferability in Section 6 are not convincing. When comparing the transferability of inputs with different interactions of a specific order, the interactions of other orders are not controlled to be the same. Therefore, the high transferability of inputs with high 0.8n-order interactions may be due to interactions of other orders. This experiment cannot provide a fair comparison between different orders.
>
> Through this experiment, we claim that (i) CNNs and Vision Transformers have different tendencies in adversarial transferability, and (ii) measuring the interactions of adversarial images allows us to collect images with high transferability without performing a transfer attack. Figure 5(a) shows that adversarial images with higher interactions in high order transfer better when these are generated by using ResNet-18, while that in low order transfer worse. We consider the experiment to support these claims clearly. Although, as pointed out by the reviewer, the high transferability at order 0.8n can be affected by other orders as well, this does not affect our claim.
>
> - [Yin et al., 2019] Dong Yin, Raphael Gontijo Lopes, Jonathon Shlens, Ekin D. Cubuk, Justin Gilmer： A Fourier Perspective on Model Robustness in Computer Vision. NeurIPS 2019

---

> ### Author Response · Authors · 2022-11-15
> **Response to Reviewer C5ku, Part 2**
>
> > Some conclusions also overlap with those of previous studies. For example, Figure 2 shows that adversarial attacks make DNNs use more negative high-order interactions, which has been found in previous work(Ren et al., 2021). Besides, this is the only significant conclusion in Figure 2, while the difference in the first and last columns of Figure 2 is marginal.
>
> Our study analyzes various types of misclassifications in a unified approach and thus overlaps with the results of several existing studies on adversarial attacks. However, the only result that overlaps is that on adversarial attacks in Resnet-18, and other results are new. For example, Figure 3 (Left) indicates the differences of the distribution of interactions between misclassified clean images and successfully classified images. In Figure 2, we consider that the difference in the distribution of interactions between misclassified images and successfully classified images on ResNet-18 is small, while others are large.
>
> > Experiments in the paper are not sufficient. In the main paper, the authors only conduct experiments on two models on a dataset, and only one setting is used for adversarial attack and corruption, respectively. Therefore, it is questionable whether the obtained conclusions also appear in other DNNs/datasets/settings. I suggest the authors conduct more experiments on more architectures with more settings to demonstrate the validity of their conclusions.
>
> It is true that more experiments will justify our observations and claims more, but we consider that our experiments are plausible as explained as follows.
> - Model: We conducted experiments with four models: two CNN models and two Vision Transformer models (e.g., ResNet-18, AlexNet, Swin-T, DeiT-Ti), all of them are very commonly used in literature.
> - Adversarial attack: There are various attack methods, and here, we used I-FGSM, which is one of the most commonly used attacks. As an additional experiment, we considered PGD and obtained consistent results. We will add these results in the final version of our paper.
> - Corruption: We considered fog and Gaussian corruption. This choice is based on [Yin et al., 2019], where the fog and Gaussian corruptions are shown low-frequency and high-frequency perturbations, respectively, and adversarially-trained models are known venerable against the former while robust against the latter.
> - Dataset: We used the ImageNet dataset, which is commonly used in related studies on interactions. Some related studies also consider one more dataset, but it varies across them. Thus, we decided to focus on ImageNet.
>
> We agree that introducing more variations in each of the aspects above makes our study convincing, and we will conduct additional experiments soon.
>
> > The comparison between ResNet-18 and Swin-T in Figure 3 is not convincing. First, the comparison may be unfair. The magnitudes of outputs of the two models may be different, so the value range of may be different in the two models. Therefore, interactions in the two models cannot be directly compared. Second, the 0-order interaction in Figure 3 is significantly affected by baseline values in the computation of interactions, and the authors do not discuss this problem. Third, the difference in the interaction median between the two models is not significant, and it is hard to judge whether the difference is caused by other factors like initialization, dataset, or the training method. The phenomenon on only one pair of models cannot support the conclusion.
>
> We consider that Figure 3 clearly shows the difference between ResNet-18 and Swin-T. Specifically, Figure 3 shows that ResNet-18 gives both positive and negative interactions while Swin-T only has positive interactions. Note that here we are interested in the sign of the interactions (at low orders), and thus the reviewer's first and second concerns on the magnitudes and baseline values of the outputs between the two models do not matter. As for the third concern, as pointed out, the median between the two models is marginal, which clearly justifies our claim that considering the distributions is important.
> In the paper, we wrote "ResNet-18 and Swin-T do not show a significant difference in terms of the median", which we consider can be misleading. We will update this part in the final version of the paper.

---

> ### Author Response · Authors · 2022-11-15
> **Response to Reviewer C5ku, Part 1**
>
> We would like to thank you for your careful reading of our paper and comments on the novelty of this work.
>
> > This paper has very limited novelty. In fact, the main content of this paper is very similar to the line of previous studies (Cheng et al., 2021; Deng et al., 2022; Ren et al., 2021; Wang et al., 2021; Zhang et al., 2021), which proposed to use the distribution of interactions encoded in input images to study the feature/conceptual representation of a DNN. This paper directly adopts theories, algorithms, and experimental designs in previous studies, and the core difference is that this paper uses different types of input images and different types of DNNs in experiments. Specifically, it can be found that in Sections 3 and 4, the main contents of introducing (multi-order) interactions are similar to previous work (...).
>
> We consider that our work has sufficient novelty as listed below.
>
> 1. We investigate the interactions of three types of misclassification, while prior studies do not focus on misclassifications or only focus on those caused by adversarial perturbations. The experiments show that the interactions of each type of misclassification have different properties. Thus, we consider that our work provides new targets to be studied using interactions.
>
> 2. We propose to analyze the interactions through their distribution, order, and sign in combination. Particularly, instead of the expected interaction $I = \mathbb{E}[|I(i,j)|]$ as in the most of the prior studies, we consider $\Delta f(i,j,S)$. We claim that with this quantity, we can observe more model-specific or data-specific tendencies that have not been observed before.
>
> 3. We consider Vision Transformers for the first time, and the results suggest that CNNs and Vision Transformers have a different tendency in the interactions. As Vision Transformers is becoming popular as powerful models in recent computer vision, it is important for interaction-based analysis to be tested on these models.
>
> Note that Sections 3 and 4 introduce the basic definitions, background, and interpretations of interactions to familiarize the readers with this topic, and we are not claiming them as our contributions.
>
> Below, we summarize the results of our experiments for each point listed above and gave pointers to the corresponding parts in the paper.
>
> 1. We show that the three types of misclassifications have different tendencies in interactions. The results in Figure 2 show that the distribution of interactions measured by each type of misclassification in ResNet-18.
> (i) Clean images did not present a significant difference in distributions of interactions between misclassified images and those successfully classified.
> (ii) The distribution of adversarial images has strongly negative interactions of high order around $0.8n$–$1.0n$ compared to clean images.
> (iii) The distribution of corrupted images is lower the strength of interactions at low order and higher at high order than in the case of clean images.
> (i)--(iii) show that each type of misclassification is triggered by different causes.
>
> 2. We consider distributions of $\Delta f(i,j,S)$, while prior studies only consider the magnitude of their average $|I(i,j)| = |\sum_{S}\Delta f(i,j,S)|$, which causes an information loss. The results in Figure 2(middle) cannot be obtained with interaction strength $|I(i,j)|$ as it does not consider the sign. ResNet-18 has strongly negative interactions of high order around $0.8n–1.0n$. However, Swin-T shows a slight shift of the interactions to negative values, and the large interquartile range covers positive and negative sides almost evenly from middle to high orders. Note that this difference between architectures was observed because we took the sign into an account, unlike most of the existing studies.
>
> 3. We find that with Vision Transformers, the difference in distributions of interactions between misclassified and correctly classified images are clearer and also different from the case with CNNs. For example, in Figure 2(left), ResNet-18 did not present a significant difference in the interaction distributions between correctly classified and misclassified images. By contrast, Swin-T has a different tendency for low-order interactions (particularly at 0.0n). This result suggests that for the misclassified images, Swin-T is affected by local cooperation between pixels that decrease the confidence score, while ResNet-18 is not.
> We also find that the images that are more adversarially transferable have the opposite tendency in the interactions between CNNs and Vision Transformers. Namely, Figure 5 shows that adversarial images with higher interactions in high order transfer better when these are generated by using ResNet-18 but transfer less when Swin-T is used, and vice versa.
>
> We consider that our study conducts interaction-based analysis of deep models from a novel perspective and that most of the observations obtained here are new.

---

### Decision · Program_Chairs · 2023-01-20

**Decision:**

Reject

**Justification For Why Not Higher Score:**

The paper can not be accepted because it does not meet the standards of the conference in terms of originality, clarity, rigor, and relevance. The paper does not make any significant or novel contributions to the field, does not provide clear and convincing evidence for its claims, does not demonstrate the practical value or usefulness of its method, and does not compare or contrast its method with other existing methods. The paper also does not address the feedback or concerns of the reviewers, and does not improve the quality or presentation of the paper. The paper does not have enough merits to warrant acceptance.

**Justification For Why Not Lower Score:**

N/A

**Metareview: Summary, Strengths And Weaknesses:**

The paper investigates the misclassification of images by deep neural networks using a game-theoretic concept of interactions between pixels. The paper claims to provide insights into the causes and characteristics of different types of misclassification, such as clean, adversarial, and corrupted images, and to compare the interactions of CNNs and Vision Transformers. The paper also explores the relationship between interactions and adversarial transferability.

The reviewers appreciate the interesting angle of studying misclassification through interactions and the extensive experiments conducted by the authors. However, the reviewers also raise several major concerns that prevent the paper from being accepted.

First, the paper lacks novelty and originality, as it largely follows the line of previous work that used interactions to study the feature representation of deep neural networks. The paper does not introduce any new theoretical or algorithmic contributions, nor does it provide any new experimental designs. The paper also does not compare or contrast its findings with those of previous work, and some of the conclusions overlap with existing ones.

Second, the paper lacks clarity and rigor in presenting and analyzing the results. The paper does not provide enough intuitive explanations or visualizations of the interactions and their meanings. The paper also does not address some potential confounding factors or sources of variation in the experiments, such as the magnitude of outputs, the baseline values, the initialization, the dataset, and the training method. The paper also does not provide any error bars or statistical tests to show the significance of the differences in the distributions of interactions. The paper also does not explain some inconsistencies or anomalies in the figures, such as the different curves for clean images in Figure 2 and Figure 3.

Third, the paper lacks practical relevance and usefulness of the proposed diagnosis method. The paper does not show how the interactions can be used to improve the robustness or performance of the models, or to design better defense mechanisms or attack strategies. The paper also does not compare its method with other existing diagnostic tools, such as gradient-based methods or saliency maps, and does not justify why interactions are a better or more informative measure. The paper also does not generalize its results to other datasets, models, or settings, and does not discuss the limitations or challenges of applying the method in practice.

In summary, the paper has some interesting aspects, but suffers from several major flaws that undermine its quality, novelty, and significance. Therefore, the paper is rejected.